# Antibacterial Potential of *Bacopa monnieri* (L.) Wettst. and Its Bioactive Molecules against Uropathogens—An In Silico Study to Identify Potential Lead Molecule(s) for the Development of New Drugs to Treat Urinary Tract Infections

**DOI:** 10.3390/molecules27154971

**Published:** 2022-08-05

**Authors:** Jyoti Mehta, Kumar Utkarsh, Shivkanya Fuloria, Tejpal Singh, Mahendran Sekar, Deeksha Salaria, Rajan Rolta, M. Yasmin Begum, Siew Hua Gan, Nur Najihah Izzati Mat Rani, Kumarappan Chidambaram, Vetriselvan Subramaniyan, Kathiresan V. Sathasivam, Pei Teng Lum, Subasini Uthirapathy, Olatomide A. Fadare, Oladoja Awofisayo, Neeraj Kumar Fuloria

**Affiliations:** 1Faculty of Applied sciences and Biotechnology, Shoolini University, Solan 173212, India; 2DNA Lab’s Center for Applied Sciences, Dehradun 248001, Uttarakhand, India; 3Faculty of Pharmacy, AIMST University, Bedong 08100, Kedah, Malaysia; 4Department of Pharmaceutical Chemistry, Faculty of Pharmacy and Health Sciences, Royal College of Medicine Perak, Universiti Kuala Lumpur, Ipoh 30450, Perak, Malaysia; 5Department of Pharmaceutics, College of Pharmacy, King Khalid University, Abha 61421, Saudi Arabia; 6School of Pharmacy, Monash University Malaysia, Bandar Sunway 47500, Selangor, Malaysia; 7Faculty of Pharmacy and Health Sciences, Royal College of Medicine Perak, Universiti Kuala Lumpur, Ipoh 30450, Perak, Malaysia; 8Department of Pharmacology, College of Pharmacy, King Khalid University, Abha 62529, Saudi Arabia; 9Faculty of Medicine, Bioscience and Nursing, MAHSA University, Jalan SP 2, Bandar Saujana Putra, Jenjarom 42610, Selangor, Malaysia; 10Faculty of Applied Sciences, AIMST University, Bedong 08100, Kedah, Malaysia; 11Department of Pharmacology, Faculty of Pharmacy, Tishk International University, Erbil 44001, Kurdistan Region, Iraq; 12Organic Chemistry Research Lab, Obafemi Awolowo University, Osun 220282, Nigeria; 13Department of Pharmaceutical and Medical Chemistry, University of Uyo, Uyo 520003, Nigeria; 14Center for Transdisciplinary Research, Department of Pharmacology, Saveetha Dental College and Hospital, Saveetha Institute of Medical and Technical Sciences, Saveetha University, Chennai 600077, India

**Keywords:** *Bacopa monnieri*, antimicrobial, agar well diffusion, molecular docking, MD simulation, urinary tract infection

## Abstract

Urinary tract infections (UTIs) are becoming more common, requiring extensive protection from antimicrobials. The global expansion of multi-drug resistance uropathogens in the past decade emphasizes the necessity of newer antibiotic treatments and prevention strategies for UTIs. Medicinal plants have wide therapeutic applications in both the prevention and management of many ailments. *Bacopa monnieri* is a medicinal plant that is found in the warmer and wetlands regions of the world. It has been used in Ayurvedic systems for centuries. The present study aimed to investigate the antibacterial potential of the extract of *B. monnieri* leaves and its bioactive molecules against UTIs that are caused by *Klebsiella pneumoniae* and *Proteus mirabilis*. This in vitro experimental study was conducted by an agar well diffusion method to evaluate the antimicrobial effect of 80% methanol, 96% ethanol, and aqueous extracts of *B. monnieri* leaves on uropathogens. Then, further screening of their phytochemicals was carried out using standard methods. To validate the bioactive molecules and the microbe interactions, AutoDock Vina software was used for molecular docking with the *Klebsiella pneumoniae* fosfomycin resistance protein (5WEW) and the Zn-dependent receptor-binding domain of *Proteus mirabilis* MR/P fimbrial adhesin MrpH (6Y4F). Toxicity prediction and drug likeness were predicted using ProTox-II and Molinspiration, respectively. A molecular dynamics (MD) simulation was carried out to study the protein ligand complexes. The methanolic leaves extract of *B. monnieri* revealed a 22.3 mm ± 0.6 mm to 25.0 mm ± 0.5 mm inhibition zone, while ethanolic extract seemed to produce 19.3 mm ± 0.8 mm to 23.0 mm ± 0.4 mm inhibition zones against *K. pneumoniae* with the use of increasing concentrations. In the case of *P. mirabilis* activity, the methanolic extracts showed a 21.0 mm ± 0.8 mm to 24.0 mm ± 0.6 mm zone of inhibition and the ethanol extract produced a 17.0 mm ± 0.9 mm to 23.0 mm ± 0.7 mm inhibition zone with increasing concentrations. Carbohydrates, flavonoids, saponin, phenolic, and terpenoid were common phytoconstituents identified in *B. monnieri* extracts. Oroxindin showed the best interactions with the binding energies with 5WEW and 6Y4F, −7.5 kcal/mol and −7.4 kcal/mol, respectively. Oroxindin, a bioactive molecule, followed Lipinski’s rule of five and exhibited stability in the MD simulation. The overall results suggest that Oroxindin from *B. monnieri* can be a potent inhibitor for the effective killing of *K. pneumoniae* and *P. mirabilis*. Additionally, its safety has been established, indicating its potential for future drug discovery and development in the treatment for UTIs.

## 1. Introduction

Urinary tract infections (UTIs) are among the most common bacterial diseases for women and the elderly. Although they may result in illnesses that are not life-threatening, affected patients can face extreme distress [1]. UTIs are caused by a variety of organisms, including *Escherichia (E.) coli, Klebsiella (K.) pneumoniae, Enterococcus (E.) faecalis, Proteus (P.) mirabilis*, *Staphylococcus (S.) saprophyticus*, *Pseudomonas (P.) aeruginosa, Enterobacter* species, *Streptococcus* species, and *S. aureus* [2]. Initially, infections are usually caused by a single species of bacteria, such as the uropathogenic *E. coli* or *E. faecalis* [3]. However, over time, many organisms, including *K. pneumoniae*, *P. aeruginosa*, *P. mirabilis*, and *Morganella (M.) morganii*, can colonize the urinary tract and form polymicrobial biofilms [4,5]. Both *P. mirabilis* and *K. pneumoniae* are Gram-negative bacteria that live in human fecal flora as harmless commensal bacteria, inhabiting the gastrointestinal tract. They are, however, known to cause a wide range of opportunistic human illnesses, particularly wound infections, respiratory tract infections, and UTIs [6,7].

UTIs predominantly affect women when the bacteria infect any part of the urinary system, including the kidneys, the bladder, the uterus, or the urethra. Pregnancy, a history of an earlier UTI, age, sexual behavior, and lack of hygiene are among the risk factors for UTIs. A significant prevalence of UTIs exists across all socioeconomic categories, exacerbating economic pressure for some families. UTIs are one of the most prevalent healthcare-associated infections, with an estimated 150 million people infected worldwide each year, with 13,000 associated deaths [8]. Although many antibiotics are prescribed for the treatment of UTIs, a major cause for concern is the emergence of drug-resistant strains.

Because microbial infections tend to be more severe in immune-deficient patients, there is an urgent need for alternative therapies, including natural products, to ameliorate multidrug resistances (MDRs) that are commonly seen. The rapid, widespread emergence of resistance to newly introduced antimicrobial agents indicates that even new families of antimicrobial agents have a short life expectancy. Natural products have numerous advantages, including fewer side effects, improvements in patient tolerance, lower costs, widespread acceptance due to a long history of usage, and sustainability. Many essential drugs have been discovered and developed as a result of phytochemical and pharmacological studies of natural products [9]. Historically, medicinal plants deliver a wide variety of components with confirmed therapeutic qualities [10]. The renewed interest in plant-derived therapies seems to be due primarily to the widely held assumption that “green medicine is safe.” 

The consumption of antiseptic and anti-adhesive herbs, including the leaf parts of *Arctostaphylos uva-ursi* (Uva ursi) and *Juniperus* spp. (Juniper) and the fruit part of *Vaccinium macrocarpon* (cranberry), are able to excrete antimicrobial constituents, which may either kill microbes directly or interfere with their attachment to epithelial cells, thereby preventing acute as well as chronic UTIs [11]. Berberine extracted from *Mahonia aquifolium* (Oregon grape) and *Hydrastis canadensis* (Goldenseal) is found to be an effective drug in combating infections that are caused by *E. coli* and *Proteus* species by hindering their adherence to the host cell [12], suggesting their potent role in the treatment of UTIs.

*Bacopa monnieri* (L.) Wettst. *(B. monnieri),* popularly known as Brahmi or water hyssop, is one of the perennial creeping plants found in various parts of the world. It is also known as “the herb of grace,” due to its various medicinal properties and memory-enhancing effects [13,14]. It belongs to the family Scrophulariaceae that grows optimally in wet, damp, and marshy areas, with succulent, oblong leaves and small white flowers as its descriptive features. The *Bacopa* species is a native herb that grows in the wetlands of southern and eastern India, Europe, Australia, Africa, Asia, North America, and South America. In addition to being known as a nootropic herb, the bacopa plant aids in the regeneration of injured neurons, neuronal synthesis, and synaptic activity restoration, as well as in strengthening brain activity.

*B. monnieri* contains brahmine, nicotinine, herpestine, bacosides A and B, saponins A, B, and C, triterpenoid saponins, stigmastanol, β-sitosterol, betulinic acid, D-mannitol, α-alanine, serine, pseudojujubogenin glycoside, aspartic and glutamic acids, and other elements [15]. For centuries, the magical herb has been used by Ayurveda medical practitioners for the treatment of various conditions, such as epilepsy, anxiety, and various stress-related disorders [16]. It also has positive effects in boosting brain function and improving memory [17]. Additionally, it is used to treat digestive complaints and skin disorders, as an antiepileptic, antipyretic, and analgesic [18,19], and as an antimicrobial agent against UTIs [20] (Figure 1).

The therapeutic activity of herbal medicines is attributed either to the individual activity or the synergistic activity of the phytomolecules. Synthetic drugs are expensive and may be inadequate for the treatment of some diseases; they are also often associated with adulterations and adverse effects. Likewise, while natural products possess a wide range of biological activities, some have toxicity issues. Therefore, computational techniques applied in drug designing have a vital role in predicting the toxicity of compounds and in determining their other properties [21]. For example, in silico studies that focus on predictions of toxicity, via ProTox-II server, can decrease the need for subsequent drug testing in animals (in vivo), while molecular docking for each phytochemical can determines its therapeutic efficiency. Such studies require less testing time, and they are economical in determining the effects of drugs on animals [22]. Hence, the aim of the present study was to explore the effects of *B. monnieri* against uropathogens, such as *K. pneumoniae* and *P. mirabilis*, using in vitro methods, while studying their active molecules via in silico approaches.

## 2. Results

### 2.1. Antimicrobial Activity of Crude Extracts of B. monnieri

Using the agar well diffusion method, the diameter of the zone of inhibition less than 11 mm was not considered and was marked as no-zone. Overall, the methanolic extract of *B. monnieri* had the highest inhibition zone against both microorganisms (*K. pneumoniae* and *P. mirabilis*) (Table 1). The effect was concentration dependent. The aqueous extract was least active against tested pathogens, while no activity was seen from the negative control, dimethyl sulfoxide (DMSO).

### 2.2. Phytochemical Screening

Based on phytochemical analysis, the methanolic and ethanolic extracts of *B. monnieri* were found to contain carbohydrates, flavonoids, tannin, saponins, steroids, phytosterols, and phenolic compounds (Table 2). The aqueous extract of *B. monnieri* indicated the presence of carbohydrates, flavonoids, saponins, steroids, and phytosterols. Tannins and phenolic compounds were absent in the aqueous extract of *B. monnieri*.

### 2.3. Molecular Docking Analysis of Major Bioactive Molecules from B. monnieri with UTI Proteins (5WEW and 6Y4F)

Molecular docking of 17 major bioactive molecules and a standard drug (Trimethoprim) with the Klebsiella pneumoniae fosfomycin resistance protein (5WEW) and the Zn-dependent receptor-binding domain of Proteus mirabilis MR/P fimbrial adhesin MrpH (6Y4F) was carried out using AutoDock Vina software to study the interaction of the bioactive molecules with the target protein. The results indicated that among all of the selected bioactive molecules—Bacopaside I (−8.4 kcal/mol), β-sitosterol (−7.7 kcal/mol), Oroxindin (−7.5 kcal/mol), and Bacoside A (−7.5 kcal/mol) with 5WEW—Bacopaside I achieved hydrogen bonding with Arg122, Tyr131, Tyr65, His115, Tyr103, Ser71, and showed hydrophobic interactions with Asp44, Phe70, His67, Leu119, and Alkyl, or π-alkyl interactions with Pro107 and Val89. β-sitosterol showed hydrophobic interactions with Gln 24, Pro 59, Pro 60, Asp 106, and Alkyl, or π-alkyl interactions with Ala 20, Leu 25, Phe 21, Val 57, Val 89, Ala 90, Leu 105, and Pro 107. Oroxindin achieved hydrogen bonding with Gly117, His115, Ser101, Tyr103, Arg122, His67, and Tyr 65, and showed hydrophobic interactions with Leu119, Ser118, Ile72, Ser71, Val116, Phe70, Ser97, and Alkyl, or π-alkyl interactions with Ala69. Bacoside A achieved hydrogen bonding with Tyr65, Lys93, Tyr131, Arg122, and Glu113, and showed hydrophobic interactions with Tyr103, Ser101, His67, Ala69, Leu45, Phe70, Leu5, Ser71, Leu119 and Alkyl, or π-alkyl interactions with Tyr68, Leu8, and His115 (Table 3 and Figure 2). In 6Y4F Luteolin (−7.5 kcal/mol), Oroxindin (−7.4 kcal/mol), and Bacopaside I (−7.3 kcal/mol) showed the best interactions. Luteolin achieved hydrogen bonding with Arg118, Thr116, and His72, and showed hydrophobic interactions with Ile140, His74, Asn82, Gly81, His117, Asn39, Ala40, and Thr4. Oroxindin achieved hydrogen bonding with Thr31 and Glu127, and showed hydrophobic interactions with Cys128, Gly153, Ile30, Ser33, Ser124, and Alkyl, or π-alkyl interactions with Ala155. Bacopaside I achieved hydrogen bonding with Ser138, Asn82, Ser137, Gln86, Ala84, and Phe85, and showed hydrophobic interactions with Glu112, Gly113, Gln114, Gly139, Pro83, and Tyr67 (Table 4 and Figure 3).

### 2.4. Drug-Likeness Predictions of Bioactive Molecules of B. monnieri

The drug-likeness properties of the major bioactive molecules from *B. monnieri* were determined using a Molinspiration online server. The drug-likeness predictions indicated that among the selected bioactive molecules, Apigenin, Nicotine, Loliolide, and Wogonin followed all the rules of drug likeness, while Rosavin, Oroxindin, Stigmastanol, β-sitosterol, Betulinic acid, Brahmic acid, Cucurbitacin B, D-mannitol, and Monnierasides I showed only one violation, which was also acceptable (Table 5), suggesting that they are phtocompounds with good binding affinities with 5WEW and 6Y4F.

### 2.5. Toxicity Predictions for Trimethoprim and the Major Bioactive Molecules from B. monnieri

The toxicity of the major bioactive molecules from *B. monnieri* was predicted using a Protox II server. Most of the selected bioactive molecules overcame all toxicity barriers in terms of binding energy or drug-likeness activity (Table 6). However, Oroxindin was considered as a potential bioactive molecule, and it was selected for further investigation as a promising future candidate for drug discovery and development.

### 2.6. Molecular Dynamics (MD) Simulation of Best Protein Ligand Complexes

The MD simulation of Oroxindin in complex with 5WEW and 6Y4F was carried out for 100 ns using GROMACS 2018.3 software to study the stability of protein ligand complexes. The MD simulation data showed that the RMSD of Oroxindin in complex with 5WEW and 6Y4F was stable from the start of the simulation and remained stable for up to 100 ns, as shown in Figure 4A. This showed that Oroxindin is competent to bind to the target protein’s binding pocket. At the same time, the RMSD showed some acceptable fluctuation between 0.5 A and 0.8 Å. Similarly, a RMSF was carried out to study the flexibility of the protein ligand complex. The RMSF plot showed that Oroxindin in complex with 5WEW had strong interactions, with 10–50 and 80–110 residues (Figure 4B), whereas Oroxindin in complex with 6Y4F showed strong interactions with residues of 25–160 (Figure 4B).

The radius of gyration for Oroxindin in complex with 5WEW was 1.7 nm and for 6Y4F it was 1.4 nm, as shown in Figure 5A. The solvent accessible surface area for Oroxindin in complex with 5WEW was 100 nm^2^ and with 6Y4F it was 70 nm^2^ as shown in Figure 5B. Oroxindin in complex with 5WEW achieved 2 hydrogen bonds up to 35 ns and 5 to 8 hydrogen bonds between 35 ns and 100 ns (Figure 6) whereas 6Y4F achieved 2 to 4 hydrogen bonds throughout the simulation (Figure 6). The radius of gyration, the solvent accessible surface area, and the hydrogen bonding interactions confirmed the stability of Oroxindin in complex with 5WEW and 6Y4F. The binding free energy of the protein ligand complexes is shown in Figure 7.

## 3. Discussion

Infectious diseases are a primary source of morbidity and mortality, and the number of MDR strains is on the rise, as is the development of strains with reduced antibiotic sensitivity. Currently, traditional medical systems are still society’s most valuable assets in dealing with a variety of ailments. Plant-based traditional medicines are widely used for a range of disorders, as they are effective as therapies and have no negative effects in patients, such as the demonstrated effects of synthetic pharmaceuticals in COVID-19 patients [23]. Pathogenic bacteria are thought to develop antibiotic resistance, making the hunt for novel antibiotics a never-ending process [24]. For these reasons, researchers have recently begun to concentrate their efforts on herbal products in order to develop more effective drugs and therapeutic targets against lethal MDR microbial strains [23,25,26,27]. These initiatives inspired us to perform the current study, which triggered the search for new antibacterial compounds generated from the methanolic and ethanolic extracts of *B. monnieri* that have been found to be effective against the bacteria that cause UTIs.

Several researchers have investigated *B. monnieri’s* antibacterial properties, but the majority of these studies have focused on standard bacterial strains [19,28,29]. Although clinical isolates of human pathogenic *E. coli* and *K. pneumoniae* have been evaluated, those strains are not MDR strains [30]. All of the tested *B. monnieri* extracts demonstrated antibacterial action against the pathogens. The methanolic extract of *B. monnieri* was found to have a stronger inhibitory effect against *K. pneumoniae* and *S. aureus* [31,32], which is consistent with our findings. Furthermore, the findings indicating that methanolic extracts are more powerful than aqueous extracts against bacterial strains are consistent with the findings of other studies [33,34,35].

The tested microorganisms were not inhibited by an aqueous extract at a concentration of 100 mg/mL, which could be due to the loss of some bioactive components during the sample extraction process. Furthermore, multiple studies have found that the type of solvent used for extraction plays a significant impact in defining the extract’s potential [36,37]. This is related to the fact that various phytochemicals have different relative solubilities in different polarities of solvents. The systemic phytoconstituent screening of plant extracts is an important technique for finding new medicinal lead compounds [38,39]. The antibacterial activities of phytochemicals are achieved through two basic pathways: direct bacterial death or the prevention of microbial adherence to epithelial cells [11]. Some of the extracts can be developed into medications or used as blueprints for drug development, due to the presence of various phytochemicals and antioxidants [40]. 

Antibacterial resistance has become one of the world’s most serious public health challenges in the last two decades [41,42]. Hence, there has been a growing interest in identifying and developing novel antimicrobial compounds from a variety of sources to tackle microbial resistance [43]. Herbal medicines such as turmeric and its bioactive molecule curcumin, have been shown to improve the symptoms of chronic UTIs, protect renal tubular function, and reduce inflammatory responses [44,45]. Herbal remedies have been demonstrated to be effective in the treatment of UTIs [46].

The present study confirmed that both the ethanolic and methanolic extracts of *B. monnieri* provide inhibitory effects against the Gram-negative bacteria *K. pneumoniae* and *P. mirabilis*. Different studies reported that *B. monnieri* extract can inhibit Gram-negative as well as Gram-positive bacteria [47]. Hema et al. [30] reported that *K. pneumoniae* and *P. vulgaris* were resistant to the ethanolic extract of *B. monnieri*, a finding that is contrary to those of many others, including ours. The present work on *B. monnieri* also showed that the crude methanolic extract was more effective than the ethanolic or aqueous extracts.

More investigations are required on different species of the bacteria that cause UTIs to provide a clearer indication for *B. monnieri* as a potential agent against UTIs. Based on the present work, it is plausible that a contributory effect against UTIs is provided by the alkaloids, tannins, and flavonoids, acting individually or in combination. Similarly, the presence of all major phytoconstituents, such as alkaloids, flavonoids, steroids, and saponins (except tannin) was investigated in the methanolic extract of *B. monnieri*. It was reported that different phytochemicals are responsible for antibacterial, cytotoxic, analgesic, and neuropharmacological activities [48]. Among the chemical constituents, alkaloids and flavonoids may be responsible for antibacterial properties [49].

Recently, many plants have been explored via in silico and in vitro methods to evaluate the antibacterial activities of their major bioactive molecules. For example, the bioactive molecules thymol and emodin of *Thymus serpyllum* and *Rheum emodi*, respectively, were found to reveal antimicrobial activity with the penicillin binding protein, together with drug likeness and toxicity prediction [50,51,52,53]. Similarly, a study carried out by Ramasamy et al. [54] revealed that aglycones and their derivatives of *B. monnieri* have better binding affinity and good central nervous system drug-like properties; they validated in silico receptor and acetylcholinesterase (AChE) models to predict the potential of bacosides and aglycones and their derivatives. The alcoholic extract was analyzed via an invivo study and was found to increase the memory of rats; the activity is attributed to saponin mixtures comprising the bacosides A and B [55]. Emran et al. [56] used via docking studies to show that luteolin, a phytochemical of *B. monnieri*, has potential to act against *Staphylococcus aureus*, as it has the highest fitness score and greater specificity toward the DNA gyrase binding site than the penicillin binding protein.

No study has been conducted to date to screen the major phytochemicals of *B. monnieri* against uropathogens. This is the first report that shows that the Oroxindin component, a flavone or a type of phenolic of *B. monnieri*, is a potent inhibitor of UTIs, establishing pathogens such as *K. pneumoniae* and *P. mirabilis*. It is yet confirmed that the Oroxindin component is safe for future use against UTIs. Further research is necessary to validate the full spectrum of efficacy of the Oroxindin antibacterial compounds from *B. monnieri* via in vitro or in vivo approaches.

Additionally, nanotechnology could be used to improve the pharmacokinetics and bioavailability of phytochemicals. Several issues that are commonly associated with the delivery of free phytochemicals, such as the rapid elimination of phytochemicals from the bloodstream, their limited absorption, and their bioavailability, reduce their potency and may necessitate greater dosages to achieve efficacy [57]. Some natural compounds are linked with undesirable side effects, such as toxicity; the use of a drug carrier may provide several benefits, including shielding the compounds from degradation in the body [58]. Nanocarriers can be altered to provide smart delivery mechanisms by using pH or temperature-sensitive materials [59], targeting ligands to improve cell uptake and selectivity, and using cellular membranes for cloaking to escape recognition by the reticuloendothelial system (RES) [60]. Several nanoparticles, liposomes, inorganic nanoparticles, and emulsions have been shown to be effective transporters for phytochemicals such as curcumin [61,62]. With respect to the future prospects of encapsulating *B. monnieri*, polymeric nanoparticles serve as excellent nanocarriers, due to their ability to entrap or surface-adsorb active compounds. They are versatile and can be coupled with cell-penetrating peptides (CPPs), which can assist in molecular intake and absorption. CPPs can carry cargo into cells via endocytosis, which improves targeting and localization to the desired location [63]. Combining the above nanotechnology with bioactive molecules from *B. monnieri* could enhance antibacterial efficacy, increase bioavailability, and reduce toxicity, especially in UTIs (Figure 8).

## 4. Materials and Methods

### 4.1. Sample Collection of B. monnieri

*B. monnieri* was collected and authenticated by Forest Research of India (FRI), Dehradun, Uttarakhand, India, under the guidance of Dr. P.K Tamta (accession number 102) and the plant name was checked in The Plant List (http://www.theplantlist.org/, accessed on 15 October 2021). The collected samples were washed properly with normal tap water. Following washing, the leaves were clipped from the plant stem followed by rewashing 2–3 times with distilled water. Subsequently, the leaves were dried in a hot air oven for 30 min at 50 °C. The dried leaves were then grounded to fine powders, using mortar and pestle, before finally being stored in airtight containers.

### 4.2. Preparation of Aqueous Extracts of B. monnieri

The powdered plant material (30 g) was added to 300 mL of distilled water in a conical flask. Then, the mouth of the conical flask was covered. The flask was kept in an orbital shaker for 24 hrs and underwent a continuous agitation at 200 rpm to ensure a thorough mixing. The suspension was filtered using a Whatman No. 1 filter paper under aseptic conditions [64]. The filtrate was collected before being concentrated by evaporation at 40 °C on a water bath. The yielded crude extract was stored at 4 °C before further use.

### 4.3. Preparation of Methanolic and Ethanolic Extracts of B. monnieri

The dried powder (30 g) was thoroughly mixed with 300 mL of organic solvents (one in 96% ethanol and another in 80% methanol) in a conical flask for their respective extract preparation. The mixture was placed at room temperature on a shaker at 200 rpm. After 48 hrs, the mixture was filtered through a sterilized Whatman No. 1 filter paper. The plant residue was re-extracted with the addition of the same solvent as before, followed by a refiltration step after 24 hrs, to yield the final concentrated slurry. The combined filtrate that was yielded was further concentrated by complete solvent evaporation at 40 °C on a water bath to yield the crude extract. Finally, the extract was checked for sterility on nutrient agar plates before being stored at 4 °C in a refrigerator for further use [65]. The dry weight of the extracts obtained by solvent evaporation was used to determine the concentration (in mg/mL) [66]. Then, stock solutions of the crude extracts for each type of organic solvent were prepared by mixing well the appropriate amount of dried extracts with 5% DMSO solvent to obtain a final concentration of 100 mg/mL.

### 4.4. Antimicrobial Activity of Crude Extracts of B. monnieri

In vitro antimicrobial activity was investigated on the methanol, ethanol, and aqueous extracts from *B. monnieri*. Urine samples from eight UTI-infected patients were collected and stored in the DNA pathology lab at the Center for Applied Sciences, Dehradun, India, with their details provided. The samples were streaked over a nutrient agar plate by a spreading method and after incubation the colony was picked and streaked again for 24 hrs to obtain a pure culture. Then, the pure culture was obtained and confirmed by Gram staining and by biochemical tests. The cultures were characterized based on their cell and colony morphology, appearance, color, colony shape, and Gram staining. The pure bacterial cultures were stained according to the 1884 methods of the Danish scientist and a physician Hans Christian Joachim Gram. The stained cells were examined with an oil immersion objective lens of a light microscope. A Gram-negative organism is characterized by a pink color. The shape of the cells was recorded. All the strains were further characterized based on biochemical tests, such as indole, methyl red, Voges-Proskauer, and citrate utilization (IMViC), as well as catalase, oxidase, carbohydrate fermentation, and urease tests to identify the organism at a species level using Bergey’s Manual of Determinative Bacteriology [67]. Both of the two microorganisms investigated were Gram-negative bacteria, i.e., *K. pneumoniae* and *P. mirabilis* (Table 7). A master plate was prepared for both organisms and maintained at 4 °C on MacConkey agar plates.

Approximately 20 mL of sterilized Mueller–Hinton agar was poured into petri dishes and allowed to solidify. Following solidification, the 24 hr nutrient broth grown with the pathogenic cultures was swabbed on the respective agar plate using sterilized cotton swabs. Subsequently, the antibacterial activity of each extract was investigated using the agar well diffusion method [68]. The concentration of the plant extracts used was 100 mg/mL. Four wells (6 mm diameter) were punched approximately 24 mm apart in the agar plates using a sterile aluminum borer. The wells were filled with 20 μL, 40 μL, and 60 μL of plant extract in three different wells, while DMSO alone was added as a negative control (for organic solvent) and distilled water (for aqueous extract) in the remaining well. Following 24 hrs of incubation at 37 °C, the plates were observed for the antibacterial activity and the diameter of the zone of inhibition for each extract was measured (in mm) with the help of a transparent plastic ruler. The experiment was performed in triplicate for each extract and the mean diameter was taken. The data represent the mean of three replicates ± standard deviation (SD).

### 4.5. Qualitative Tests for Phytochemical Analysis of Crude Extracts of B. monnieri

A phytochemical screening of extracts from different plant species was conducted to determine the presence of active secondary plant metabolites. The crude plant extracts were selected by screening, then analyzed for the presence of all major groups of bioactive molecules by employing various standard phytochemical methods [69,70,71].

#### 4.5.1. Test for Flavonoid (Alkaline Reagent Test)

Approximately 1 mL of the crude extract was treated with 5 drops of 5% sodium hydroxide (NaOH) solution, followed by the addition of 1 mL of hydrochloric acid (HCl) (2 M). An increase in the intensity of the yellow color caused it to become colorless upon the addition of a few drops of 2M HCl, which is indicative of the presence of flavonoids.

#### 4.5.2. Test for Saponins (Frothing Test)

Approximately 0.5 g of the crude extract was added to 3 mL of hot distilled water. Subsequently, the mixture was shaken vigorously for one minute. Any persistent foaming indicated the presence of saponins.

#### 4.5.3. Test for Phenols (Ferric Chloride Test)

Approximately 2 mL of the extract was treated with ethanol (2 mL) and a 5% ferric chloride (3–4 drops) solution. A deep blue color indicated the presence of phenols.

#### 4.5.4. Test for Tannins (Gelatin Test)

To a 1 mL of extract, 2 mL of 1% gelatin solution containing 10% sodium chloride was added in a test tube. The formation of white precipitate indicated the presence of tannins.

#### 4.5.5. Test for Steroids (Salkowski Test)

To a 0.5 mL of extract, 5 mL of chloroform and concentrated sulfuric acid were added. A colored layer was formed, with the upper layer turning red; a sulfuric acid layer that turned yellow was indicative of the presence of steroids.

#### 4.5.6. Test for Phytosterol (Salkowski Test)

Approximately 0.5 mL of extract was dissolved in 5 mL of chloroform and concentrated sulfuric acid along the side of the test tube. The presence of a brown ring in the middle indicated the presence of phytosterol.

### 4.6. In Silico Study of Selected Bioactive Molecules Present in B. monnieri

Open Babel GUI [72], UCSF Chimera 1.8.1, Pubchem (www.pubchem.com, accessed on 6–8 December 2021), RCSB PDB (http://www.rscb.org/pdb, accessed on 6–8 December 2021), AutoDock Vvina software [73], and Discovery Studio were used in our investigation. Docking was achieved to estimate the population of possible ligand conformations/orientations at the binding sites. To align the ligands in the same spatial coordinate, a Vina perl script was used [73]. The best conformation was selected with the minimum docked energy after the completion of the docking search. The PDB complex of protein, along with the ligands, was examined using Discovery Studio (https://discover.3ds.com/d, accessed on 6–8 December 2021) to study the interactions between the proteins and the ligands. The binding strength of the ligands was calculated as a negative score (kcal/mol).

Seventeen major bioactive molecules that are present in *B. monnieri* were selected from the 52 molecules on the basis of a literature survey [15,74,75] for the molecular docking analysis (Figure 9). The three-dimensional (3D) structures of all the bioactive molecules and the resistant trimethoprim drug were downloaded from Pubchem (www.pubchem.com, accessed on 6–8 December 2021) in .sdf format, which was further converted into a PDB file. Each selected ligand (bioactive molecules and drugs) was prepared via open Babel software, based on the command line on an Ubuntu terminal. 

The *Klebsiella pneumoniae* fosfomycin resistance protein (5WEW) and the Zn-dependent receptor-binding domain of Proteus mirabilis MR/P fimbrial adhesin MrpH (6Y4F) of the investigated microbes were used for molecular docking with the major bioactive molecules detected in *B. monnieri* to identify potential inhibitors of uropathogens, such as *K. pneumoniae* and *P. mirabilis*. The 3D structure protein (PDB ID = 5WEW & 6Y4F) was downloaded from the protein databank (http://www.rscb.org/pdb, accessed on 6–8 December 2021) as a dimer. Chain A was extracted for docking using PyMol. The active site was predicted by grid box generation (grid box dimensions = 40, 40, 40 Å) for both proteins and was centered at x, y, z = −35.853 A, 1.45 A, −20.711 Å, respectively, for 5WEW; the x, y, z values for 6Y4F were −7.534 A, −0.9 A, and 15.308 Å, respectively.

### 4.7. Drug-Likeness Calculations

The selected phytochemicals were scanned to determine the drug-likeness criteria. Lipinski’s rule of 5 using molinspiration (http://www.molinspiration.com, accessed on 6–7 December 2021) was considered to verify drug-likeness attributes, such as the number of hydrogen acceptors <10, the hydrogen donors <5, the molecular weight <500 Da, and the partition coefficient log *p* > 5. The SMILES format of all selected major phytoconstituents was uploaded for further screening [76].

### 4.8. ADMET Screening and Toxicity Prediction

Absorption, distribution, metabolism, excretion, and toxicity (ADMET) assessments were conducted to determine the absorption, toxicity, and drug-likeness properties of the major bioactive molecules selected for the study. The 3D structures of 17 bioactive molecules (Apigenin, Bacopasaponin A, Bacopaside I, β-sitosterol, Betulinic acid, Brahmic acid, Cucurbitacin B, D-mannitol, Herpestine, Loliolide, Monnierasides I, Nicotine, Oroxindin, Plantainoside B, Rosavin, Stigmastanol, and Wogonin) were saved in SMILES format and uploaded on the SWISSADME (http://www.swissadme.ch/, accessed on 8–9 December 2021) (Molecular Modeling Group of the SIB (Swiss Institute of Bioinformatics) and PROTOX-II (https://tox-new.charite.de/protox_II/, accessed on 8–9 December 2021) web tools (Charite University of Medicine, Institute for Physiology, Structural Bioinformatics Group, Berlin, Germany) [77]. SWISSADME is an online tool used to predict ADME and pharmacokinetic and physicochemical attributes of a molecule, which are the main predictors for clinical trials. Toxicity was estimated in compounds with a lethal dose of 50% (LD50) values ≤ 50 mg/kg (Class I), >50 mg/kg but < 500 mg/kg (Class II), 500 < LD50 ≤ 5000 mg/kg (Class III), and LD50 > 5000 mg/kg (Class IV). Classes I, II, and III exhibited less toxicity, whereas Class IV displayed no toxicity [78,79]. PROTOX is a rodent oral toxicity server that determines the LD50 value and the toxicity class of a target molecule [77].

### 4.9. MD Simulation of Protein Ligand Complexes

Oroxindin showed the best interactions in molecular docking and followed all the parameters for drug likeness and toxicity. To study the stability of Oroxindin in complex with 5WEW and 6Y4F, an MD simulation was carried out for 100 ns. The MD simulation was carried out using the GROMACS 2018.3 methods, as described in previous study. The solvent (using the TIP3P water model) addition was carried out in a cubic box using a box distance 1.0 nm from the closest atom in the protein. The addition of Cl^-^ ions was used to neutralize the system [80,81,82,83].

## 5. Conclusions

*K. pneumoniae* and *P. mirabilis* are sensitive against both the methanolic and ethanolic extracts of *B. monnieri*, indicating that *B. monnieri* has potential against UTIs. The extract of its leaves can confer the potential antibacterial activity due to the presence of different phytochemicals, such as alkaloids, phytosterols, saponin, phenol, flavonoids, and tannins. The present study suggests that Oroxindin, a molecule from *B. monnieri*, can be a potent inhibitor for the effective killing of *K. pneumoniae* and *P. mirabilis*. Overall, this study revealed promising in vitro and in silico results that can be used to predict efficacy in in vivo models and possibly lead to a future drug candidate. Despite the fact that we presented several observations in support of our hypothesis, we focused on a limited data set with only two bacterial pathogens. More research is needed to determine whether our findings are transferable to various uropathogens. This will assist in the identification of novel therapeutic molecules for future UTI drug discovery and development.

## Figures and Tables

**Figure 1 molecules-27-04971-f001:**
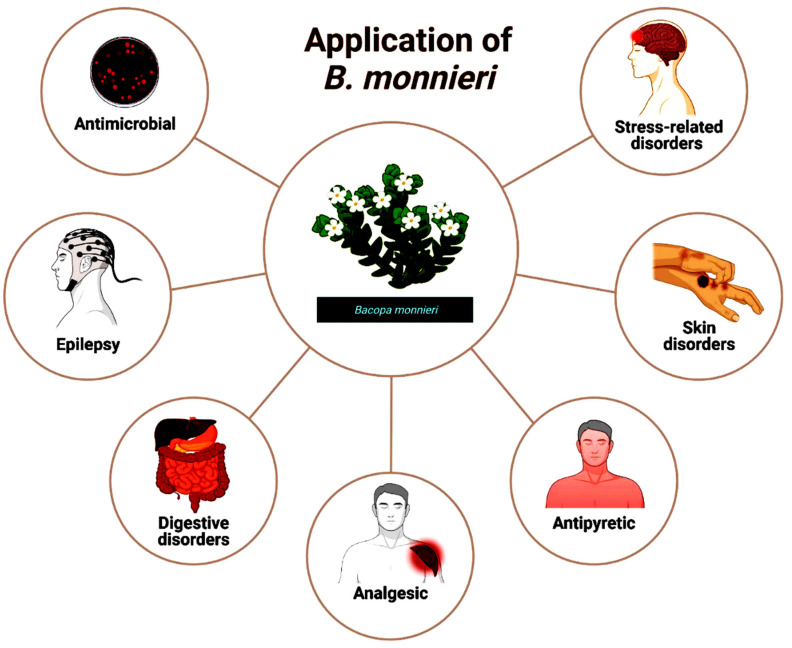
The use of *B. monnieri* to treat a variety of illnesses, including inflammations, cognitive problems, and gastrointestinal problems.

**Figure 2 molecules-27-04971-f002:**
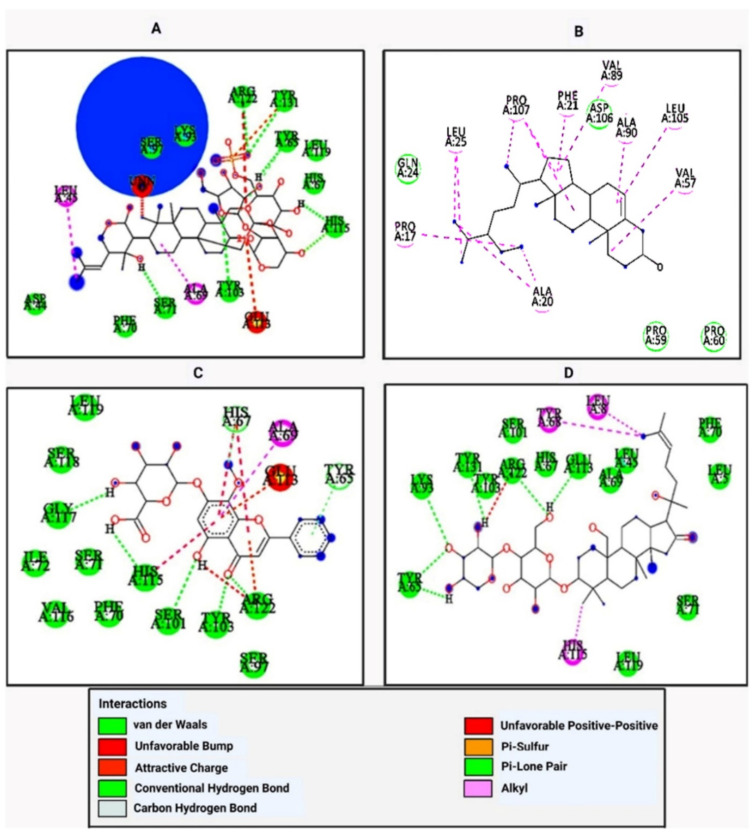
The 2D interactions of bioactive molecules of *B. monnieri* in complex with 5WEW: (**A**) Bacopaside I, (**B**) β-sitosterol, (**C**) Oroxindin, and (**D**) Bacoside A.

**Figure 3 molecules-27-04971-f003:**
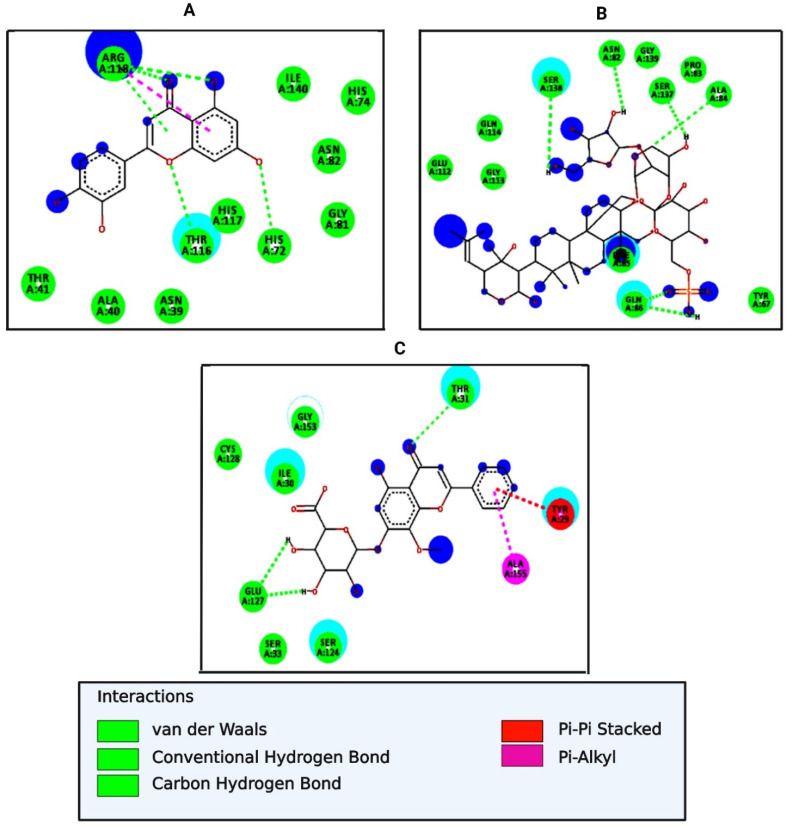
The 2D interactions of bioactive molecules of *B. monnieri* in complex with 6Y4F: (**A**) Luteolin, (**B**) Oroxindin, and (**C**) Bacopaside I.

**Figure 4 molecules-27-04971-f004:**
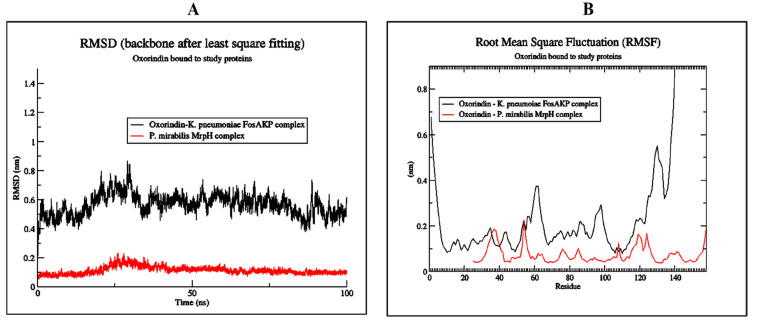
RMSD and RMSF plots of Oroxindin in complex with 5WEW and 6Y4F: (**A**) RMSD, and (**B**) RMSF.

**Figure 5 molecules-27-04971-f005:**
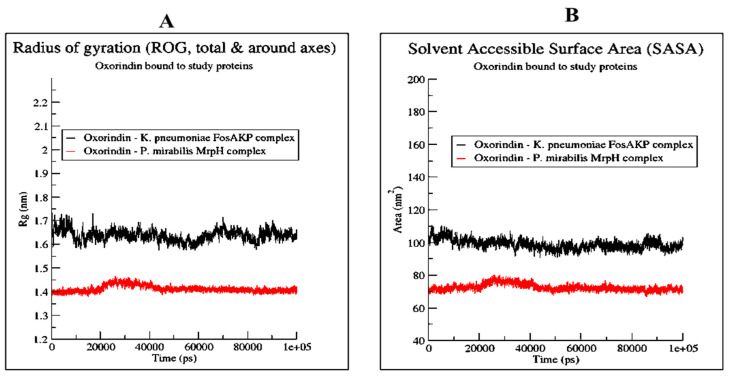
(**A**) Radius of gyration, and (**B**) solvent accessible surface area of Oroxindin in complex with 5WEW and 6Y4F.

**Figure 6 molecules-27-04971-f006:**
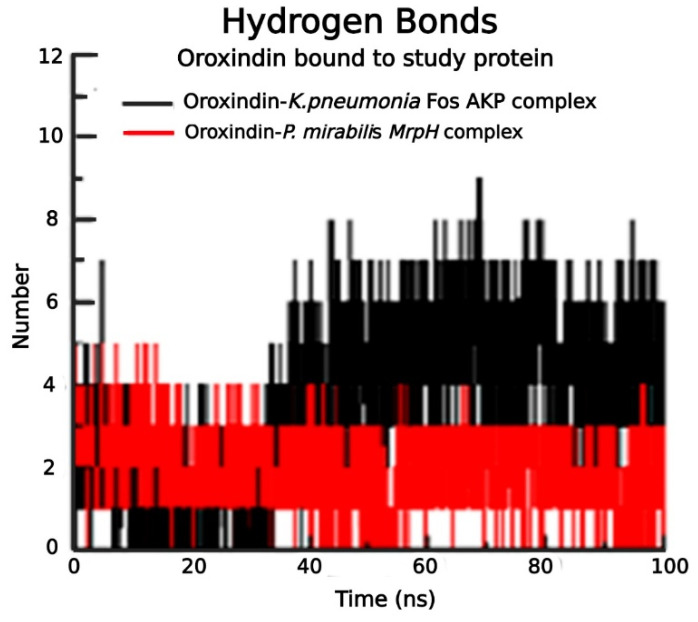
Hydrogen bond interactions of Oroxindin in complex with 5WEW and 6Y4F.

**Figure 7 molecules-27-04971-f007:**
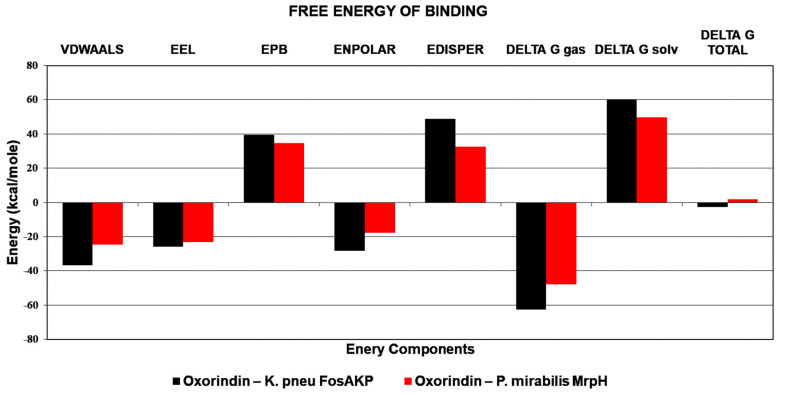
Binding free energy of Oroxindin in complex with 5WEW and 6Y4F.

**Figure 8 molecules-27-04971-f008:**
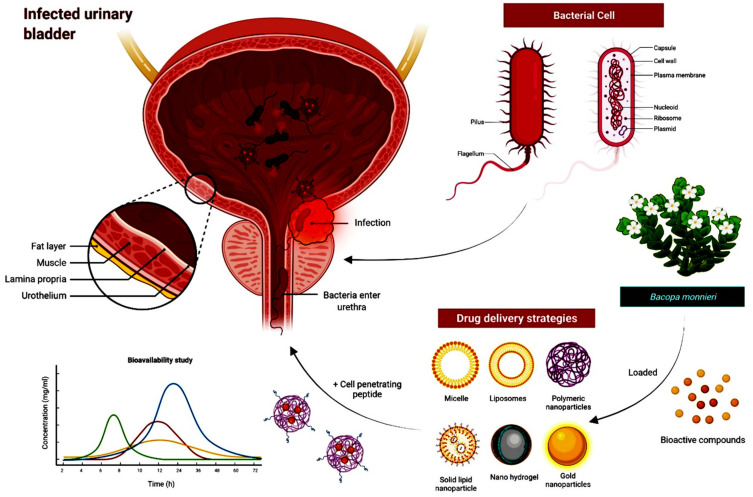
Schematic illustration of the future application of *B. monnieri* loaded inside polymeric nanoparticles enhanced with a cell penetrating peptide for improving internalization and bacterial eradication in infected bladders. The application of nanotechnology may assist in addressing toxicity concerns, while extending the half-life of bioactive compounds for effective antibacterial activities.

**Figure 9 molecules-27-04971-f009:**
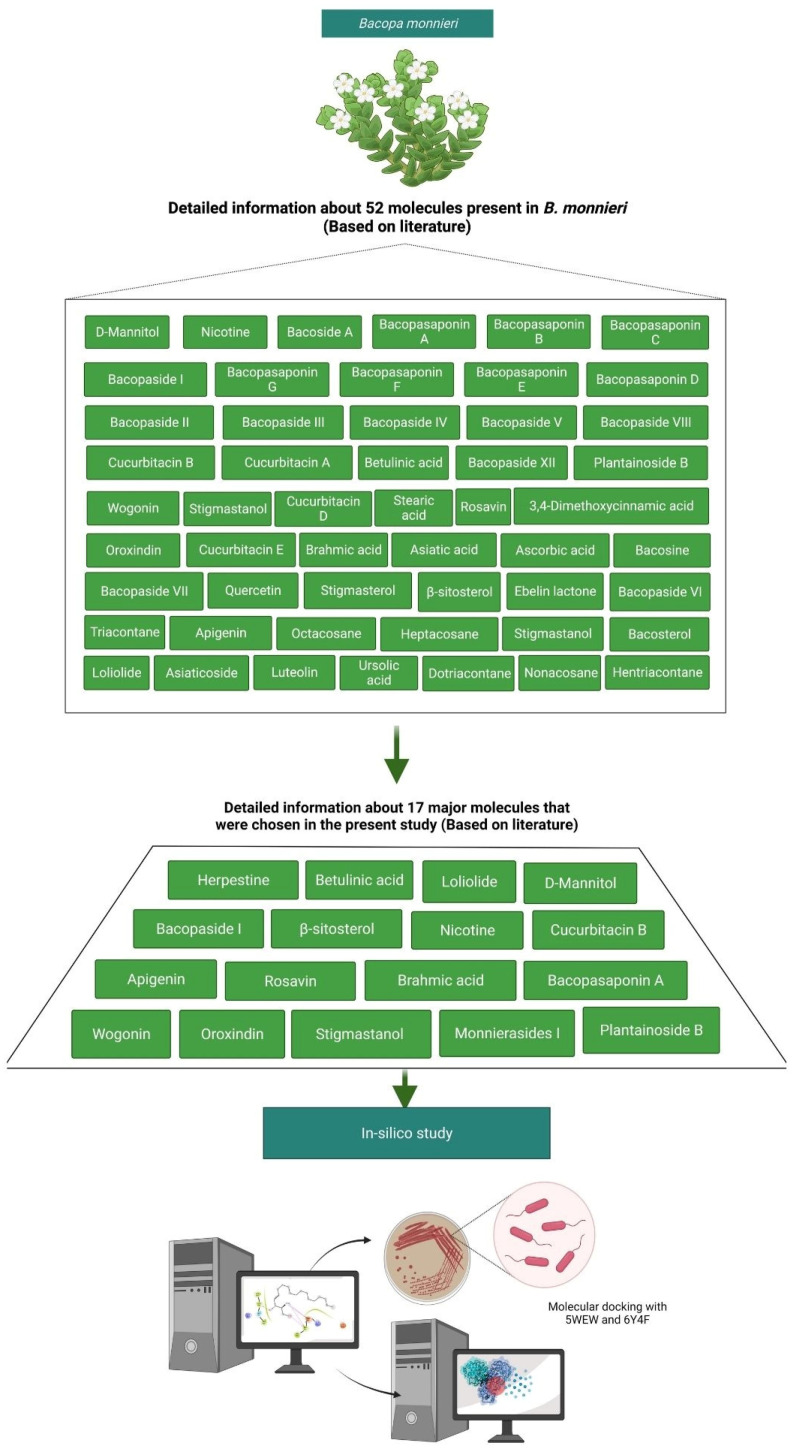
Overall process of the selected molecules from *B. monnieri* for in silico study.

**Table 1 molecules-27-04971-t001:** Antimicrobial activity of crude extracts of *B. monnieri* against *K. pneumoniae* and *P. mirabilis*.

No.	Uropathogen	Concentration (μL)	Ethanolic Extract of *B. monnieri*	Methanolic Extract of *B. monnieri*	Aqueous Extract of *B. monnieri*	Negative Control (DMSO)
		Zone of inhibition (in mm)				
1.	*K. pneumoniae*	20	19.3 ± 0.8	22.3 ± 0.6	-	-
		40	22.0 ± 0.5	24.0 ± 0.3	-	-
		60	23.0 ± 0.4	25.0 ± 0.5	11.0 ± 0.2	-
2.	*P. mirabilis*	20	17.0 ± 0.9	21.0 ± 0.8	-	-
		40	21.0 ± 1.2	21.3 ± 0.9	-	-
		60	23.0 ± 0.7	24.0 ± 0.6	-	-

(-): no zone of inhibition; DMSO = dimethyl sulfoxide; values are expressed in mean ± SD (*n* = 3).

**Table 2 molecules-27-04971-t002:** Phytochemical analysis of crude extracts of *B. monnieri*.

No.	Phytochemicals	Methanolic Extract of *B. monnieri*	Ethanolic Extract of *B. monnieri*	Aqueous Extract of *B. monnieri*
1.	Carbohydrates	+	+	+
2.	Flavonoids (Alkaline reagent test)	+	+	+
3.	Tannins (Gelatin test)	+	+	−
4.	Saponins (Frothing test)	+	+	+
5.	Steroids (Salkowski *test)*	+	+	+
6.	Phytosterols (Salkowski *test)*	+	+	+
7.	Phenolic compounds (Ferric chloride test)	+	+	−

+ Positive; − negative.

**Table 3 molecules-27-04971-t003:** Molecular docking analysis of major bioactive molecules from *B. monnieri* with 5WEW.

Bioactive Molecules/Antibiotic	Docking Score	Hydrogen Bonding	Hydrophobic Interactions	Alkyl or π-alkyl Interactions
Trimethoprim	−4.9	Lys 111, Thr 66	His 110	-
Apigenin	−6.8	-	Ala20, Gln24, Phe21, Ala87, Gly88, Leu105, Asp106	Pro107, Val89
Bacopasaponin A	−7.8			
Bacopaside I	−8.4	Arg122, Tyr131, Tyr65, His115, Tyr103, Ser71	Asp44, Phe70, His67, Leu119	Leu45, Ala69
β-sitosterol	−7.7	-	Gln 24, Pro 59, Pro 60, Asp 106	Ala 20, Leu 25, Phe 21, Val 57, Val 89, Ala 90, Leu 105, Pro 107
Betulinic acid	−7.1			
Brahmic acid	−6.9	Asp106	Leu25, Gly88, Val89, Leu105, Asp108, Gly109, Val57	Phe21, Pro107, Ala20, Pro17
Cucurbitacin B	−6.7	Thr58, Tyr68, Arg55	Ser63, Lys111, His110, Gly109, Leu10, Thr9	-
D-mannitol	−4.8	Tyr68, Arg55, Gly109, Ser63, Asp64	Thr58, Lys111, Tyr65, Thr66	-
Herpestine	−7.0	Ala69, Gly117, Ser71	Phe70, Leu45, Tyr68, His67	Leu5, Leu8
Loliolide	−5.5	Thr58, Arg55, Lys111, Thr66	Asp64, Tyr68, His67, His110, Gly109, Ser63	-
Monnierasides I	−6.9	Tyr68, Lys111, Thr58, Arg55, Arg56	His67, Thr66, His110, Gly109, Ser63, Gln54, Asp52	Ala11
Nicotine	−4.5	Tyr68, His110	Lys111, Thr66, Ser63, Thr58	
Oroxindin	−7.5	Gly117, His115, Ser101, Tyr103, Arg122, His67, Tyr65	Leu119, Ser118, Ile72, Ser71, Val116, Phe70, Ser97	Ala69
Plantainoside B	−6.8	Tyr65, Tyr103, Ser71, Leu119, Gly117, His115	Glu113, His67, Lys93, Ser118	Arg122, Ala69
Rosavin	−6.9	Gln54, Thr58, Arg56, Tyr68	Asp64, Ser63, Thr66, Gly109, Thr9, Ser50, Asp52	-
Stigmastanol	−7.0	-	Gly109, Leu105, Pro60, Asp106, Ala90, Ala20, Gln24	Pro107, Phe21, Leu25, Pro17, Val89, Ala87
Wogonin	−6.2	-	Asp44, Gly43, Ser41, His31	-
Bacoside A	−7.5	Tyr65, Lys93, Tyr131, Arg122, Glu113	Tyr103, Ser101, His67, Ala69, Leu45, Phe70, Leu5, Ser71, Leu119	Tyr68, Leu8, His115
Luteolin	−7.0	Ala69, Leu119, Tyr103	Tyr65, Lys93, His67, Ser97, Ser101, Phe70, Ser71, Gly117, Ser118	-

**Table 4 molecules-27-04971-t004:** Molecular docking analysis of major bioactive molecules from *B. monnieri* with 6Y4F.

Bioactive Molecules/Antibiotic	Docking Score	Hydrogen Bonding	Hydrophobic Interactions	Alkyl or π-alkyl Interactions
Trimethoprim	−5.1	Asn 82, Ala 84, Arg 107, Ser 137	-	-
Apigenin	−6.2	Ser124	Ile30, Asn125, Gly153, Cys152, Cys128, Tyr29, Glu32, Ser33,	-
Bacopasaponin A	−7.0	Cys152, Lys73, Tyr133, Leu146	Ile 154, Val76, Asn77, Gly78, Leu147, Pro148, Gly149, Ser150, Leu151, Asn125	Lys145
Bacopaside I	−7.3	Ser138, Asn82, Ser137, Gln86, Ala84, Phe85	Glu112, Gly113, Gln114, Gly139, Pro83, Tyr67	-
β-sitosterol	−6.5	-	Glu 32, Ser 33, Ser 124, Asn 125, Glu 127, Cys 152, Glu 153	Ile 30
Betulinic acid	−6.8	-	Arg100, Trp49, Ser108, Arg107	Arg48, Phe109, Leu104, Lys105
Brahmic acid	−6.5	Leu146	Cys152, Leu151, Ser150, Gly78, Asn77, Leu147, Pro148, Gly149, Ile154	Lys73, Lys145
Cucurbitacin B	−7.1	Gln86, Arg107, Tyr44, Phe85	Ser138, Ser137, Ala84, Thr115, Gly113, Glu112	-
D-mannitol	−5.0	Pro83, Gln86, Ser137, Ser138	Gly139, Asn82, Ala84, Phe85	-
Herpestine	−6.9	Thr116, Asn82, Arg118	Ile140, His117, Ala40, Asn39, Thr41	His72, His74
Loliolide	−5.4	Cys128, Ile30	Tyr29, Asn125, Gly153, Glu127, Cys152, Ser124, Val129	-
Monnierasides I	−7.0	Ser137, Gln86, Arg107, Ile87	Ser138, Ala84, Phe85, Ala88	-
Nicotine	−4.5	Ile30	Gly153, Ser124, Glu127, Thr31, Ser33, Glu32	-
Oroxindin	−7.4	Thr31, Glu127	Cys128, Gly153, Ile30, Ser33, Ser124	Ala155
Plantainoside B	−7.1	Ser137, Gln86, Arg107 Ile87	Tyr67, Ala88, Arg89, Phe85, Ser138	-
Rosavin	−7.1	Ser33, Gly153, Ser124, Glu127, Ile30	Thr31, Cys152, Glu32	Ala155
Stigmastanol	−6.4	-	Ser33, Glu32, Thr31, Ser124, Glu127, Gly153, Pro156, Glu47	Ile30, Tyr29, Ala155
Wogonin	−6.3	Gly149, Ser150	Leu147, Lys73, Gly78, Asn77, Tyr133	Pro59, Lys145
Bacoside A	−7.1	Gln86	Phe85, Arg107, Ile87, Asn106, Ile90	Ala88, Leu93, Val102, Arg89, Lys92
Luteolin	−7.5	Arg118, Thr116, His72	Ile140, His74, Asn82, Gly81, His117, Asn39, Ala40, Thr41	-

**Table 5 molecules-27-04971-t005:** Drug-likeness predictions of major bioactive molecules from *B. monnieri*.

Bioactive Molecules	miLog P	TPSA (Å^2^)	Number of Atoms	Number of Nitrogen and Oxygen	Number of -OH and -NHn	Number of violations	Number of Rotations	MW
Trimethoprim	0.99	105.53	21	7	4	0	5	290.32
Apigenin	2.46	90.89	20	5	3	0	1	270.24
Bacopasaponin A	3.86	176.77	52	12	6	3	5	736.94
Bacopaside I	2.54	215.00	54	13	8	3	10	768.98
β-sitosterol	8.62	20.23	30	1	1	1	6	414.72
Betulinic acid	7.04	57.53	33	3	2	1	2	456.71
Brahmic acid	3.78	118.21	36	6	5	1	2	504.71
Cucurbitacin B	2.83	138.20	40	8	3	1	6	558.71
D-mannitol	−3.10	121.37	12	6	6	1	5	182.17
Herpestine	−1.04	181.07	35	13	7	2	9	481.52
Loliolide	1.84	46.53	14	3	1	0	0	196.25
Monnierasides I	−1.11	170.05	26	10	6	1	4	370.31
Nicotine	1.09	16.13	12	2	0	0	1	162.24
Oroxindin	0.82	176.12	33	11	5	1	5	460.39
Plantainoside B	0.69	186.37	34	11	7	2	9	478.45
Rosavin	−0.95	158.30	30	10	6	1	7	428.43
Stigmastanol	8.71	20.23	30	1	1	1	6	416.73
Wogonin	2.96	79.90	21	5	2	0	2	284.27

miLogP: molinspiration LogP (to measure lipophilicity); TPSA: topological polar surface area; MW: molecular weight.

**Table 6 molecules-27-04971-t006:** Toxicity predictions of major bioactive molecules of *B. monnieri* and standard drug used for UTIs, based on ADMET SAR and Protox-II software.

Bioactive Molecules	Protox-II					
	LD_50_, (mg/kg)	Hepatotoxicity	Carcinogenicity	Immunotoxicity	Mutagenicity	Cytotoxicity
Trimethoprim	3500 (class 5)	Inactive	Active	Active	Inactive	Inactive
Apigenin	2500 (class 5)	Inactive	Inactive	Inactive	Inactive	Inactive
Bacopasaponin A	6000 (class 6)	Inactive	Inactive	Active	Inactive	Inactive
Bacopaside I	1500 (class 4)	Inactive	Inactive	Active	Inactive	Inactive
β-sitosterol	890 (Class 4)	Inactive	Inactive	Active	Inactive	Inactive
Betulinic acid	2610 (Class 5)	Inactive	Active	Active	Inactive	Inactive
Brahmic acid	2000 (Class 4)	Inactive	Inactive	Active	Inactive	Inactive
Cucurbitacin B	1190 (Class 4)	Inactive	Inactive	Active	Inactive	Inactive
D-mannitol	13500 (Class 6)	Inactive	Inactive	Inactive	Inactive	Inactive
Herpestine	500 (Class 4)	Inactive	Inactive	Inactive	Inactive	Inactive
Loliolide	34 (Class 2)	Inactive	Active	Inactive	Inactive	Inactive
Monnierasides I	190 (Class 4)	Active	Inactive	Active	Inactive	Inactive
Nicotine	3 (Class 1)	Inactive	Inactive	Active	Inactive	Inactive
Oroxindin	5000 (Class 5)	Inactive	Inactive	Active	Inactive	Inactive
Plantainoside B	5000 (Class 5)	Inactive	Inactive	Active	Inactive	Inactive
Rosavin	2000 (Class 4)	N/A	N/A	N/A	N/A	Inactive
Stigmastanol	500 (Class 4)	Inactive	Inactive	Active	Inactive	Inactive
Wogonin	1190 (Class 4)	Active	Inactive	Active	Inactive	Inactive

**Table 7 molecules-27-04971-t007:** Profile of microorganisms used in this study.

No.	Test Pathogens	Potential Infections
1	*K. pneumoniae*	Pneumonia, UTIs, septicemia, and diarrhea
2	*P. mirabilis*	UTIs, including cystitis and pyelonephritis

## Data Availability

The data presented in this study are available on request from the corresponding author.

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
