# Peer review of "Antibacterial Potential of *Bacopa monnieri* (L.) Wettst. and Its Bioactive Molecules against Uropathogens—An In Silico Study to Identify Potential Lead Molecule(s) for the Development of New Drugs to Treat Urinary Tract Infections"

_molecules, 2022, doi:10.3390/molecules27154971_

Round 1

Reviewer 1 Report

Dear authors,

The evaluated study is focused on the interesting issue of evaluating the biological effects of Bacopa monnieri and determining selected chemical parameters. Unfortunately, the name of the study does not correspond to the content and focus of the study, so I recommend changing the name significantly. Furthermore, the study is not written in a clear structure, the chapters are not properly ordered, and often the reader gets absolutely lost in what was found and what the conclusions of the study are. The structure of the entire publication needs to be changed and put together in some logical spirit according to what should emerge from the text (see the meaningless chapters at the beginning of chapter 2 - why are they presented here? What does it tell the readers? What is the purpose of the conclusion of Gram staining and biochemical tests covered?? ..... etc.....). The entire text must be reworked and revised. Additional comments on the text: 1/L. 63 and many others - why is "oroxindin" capitalized? Also on L. 283, etc. 2/ L. 69 - inappropriate capital letter at the beginning of one keyword 3/ L. 79 - the abbreviation "E." was used, but it was not explained above. 4/ The structure of the publication needs to be revised, e.g. 2.1 and 2.2 do not make sense without any explanation and relation to some results, strains, etc...? I also definitely do not consider the three-line "Gram's staining" chapter to be appropriate - connect appropriately. 5/ L. 172 - capital letter for the name of the compound DMSO, the same as L. 176, etc. 6/ L. 176 - "S.D" - incorrect entry - either without dots or the dot at the end is missing. 7/ MM section - the structure of the chapter must be revised, better organized (e.g. biological vs. chemical analysis, etc.) 8/ L. 411 - ethanol - no indication of concentration 9/ chapter 4.4 - statistics, evaluation of results? 10/ chapter 4.7.1 - as a methodology, there is no need to mention it.

Author Response

POINT BY POINT RESPONSE TO REVIEWERS

Reviewer 1

The evaluated study is focused on the interesting issue of evaluating the biological effects of Bacopa monnieri and determining selected chemical parameters. Unfortunately, the name of the study does not correspond to the content and focus of the study, so I recommend changing the name significantly.

Reply: Thank you very much for your comment. In this research, we have done antibacterial activity of Bacopa monnieri (in-vitro) and its bioactive molecules (in-silico). Based on that, we were able to identify potential lead molecule(s) that might be employed in drug discovery and development for the treatment of urinary tract infections. We have revised the title for the readers' better understanding in light of this and the reviewer comments. Following is the new title

Antibacterial potential of Bacopa monnieri (L.) Wettst. and its bioactive molecules against uropathogens – An in-silico study to identify potential lead molecules for the development of new drugs to treat urinary tract infections 

Furthermore, the study is not written in a clear structure, the chapters are not properly ordered, and often the reader gets absolutely lost in what was found and what the conclusions of the study are. The structure of the entire publication needs to be changed and put together in some logical spirit according to what should emerge from the text (see the meaningless chapters at the beginning of chapter 2 - why are they presented here? What does it tell the readers? What is the purpose of the conclusion of Gram staining and biochemical tests covered?? ..... etc.....). The entire text must be reworked and revised.

Reply: I greatly appreciate your opinion. Table 1 and sections 2.1 and 2.2 have been deleted. In the revised manuscript, all the sections has been reworked. The section and table numbers have been adjusted appropriately in both methodology and results sections. Thank you. All of your suggestions have been taken into consideration in order to improve the quality of our manuscript. All changes have been indicated in yellow in the revised version of the manuscript. The following are detailed responses to each of the reviewer comments.

No

Comments and Response

Page/Section/Location

1

L. 63 and many others - why is "oroxindin" capitalized? Also on L. 283, etc.

Since this is a prominent molecule, we have chosen to capitalise the first letter of oroxindin to prevent misunderstanding and word mixing with other words. The entire manuscript followed this format. Thank you.

Entire Manuscript

2

L. 69 - inappropriate capital letter at the beginning of one keyword

I appreciate you pointing up this format error. In the revised manuscript, the word "Agar well diffusion" has been modified to "agar well diffusion." Thank you.

Keywords

3

L. 79 - the abbreviation "E." was used, but it was not explained above.

Thank you for bringing this format problem to our attention. The phrase "Enterococcus faecalis" has been changed to "Enterococcus (E.) faecalis" in the revised manuscript. Thank you.

Introduction

4

The structure of the publication needs to be revised, e.g. 2.1 and 2.2 do not make sense without any explanation and relation to some results, strains, etc...? I also definitely do not consider the three-line "Gram's staining" chapter to be appropriate - connect appropriately.

I greatly appreciate your opinion. Table 1 and sections 2.1 and 2.2 have been deleted. In the revised manuscript, the section and table numbers have been adjusted appropriately. Thank you.

Results section

5

L. 172 - capital letter for the name of the compound DMSO, the same as L. 176, etc.

Thank you for bringing this format problem to our attention. The phrase "Dimethyl sulfoxide" has been changed to "dimethyl sulfoxide" in the revised manuscript. Thank you.

Section 2.1

6

L. 176 - "S.D" - incorrect entry - either without dots or the dot at the end is missing.

Thank you for bringing this format problem to our attention. The phrase "(mean ± S.D)" has been changed to "(mean ± SD)”in the revised manuscript. Thank you.

Table 1

7

MM section - the structure of the chapter must be revised, better organized (e.g. biological vs. chemical analysis, etc.)

The section on the in-silico/molecular docking investigation has been appropriately revised and restructured. Thank you.

Section 4.6

8

L. 411 - ethanol - no indication of concentration

The concentration of pure ethanol has been included in the revised manuscript. Thank you.

Section 4.3

9

chapter 4.4 - statistics, evaluation of results?

Reviewer makes a good point. The data represent mean of three replicates ± standard deviation (SD). This was included in the revised manuscript.

Section 4.4

10

chapter 4.7.1 - as a methodology, there is no need to mention it.

In the revised manuscript, the phrase "the method given" has been deleted, and this section has been correctly rearranged. Thank you.

Section 4.4

Reviewer 2 Report

This article describes the antibacterial activity of a medicinal plant, B. monnieri, against several bacterial strains that cause urinary tract infections. The study reveals that the antibacterial activity of the extracts depends on the utilized solvent. The bioactive molecules and their interactions with bacteria were validated through molecular docking. The article is well structured, well written, and may be of interest to pharmaceutical and biomedical domains. I recommend publishing this work, after a few minor corrections:

1. "in vivo", "in vitro" and "via" should be written in italics throughout the text;

2. "Gram-positive" and "Gram-negative" should be written with capital G.

3. The abbreviation "MD" from MD simulation should be explained where is first used in the text. 

4.  The Figures should be checked to ensure the writing is readable. For example, the color legend in Figure 2 is not readable.  

Author Response

POINT BY POINT RESPONSE TO REVIEWERS

Reviewer 2

This article describes the antibacterial activity of a medicinal plant, B. monnieri, against several bacterial strains that cause urinary tract infections. The study reveals that the antibacterial activity of the extracts depends on the utilized solvent. The bioactive molecules and their interactions with bacteria were validated through molecular docking. The article is well structured, well written, and may be of interest to pharmaceutical and biomedical domains. I recommend publishing this work, after a few minor corrections:

Thank you so much for your comments and appreciation on the manuscript's structure. All of your suggestions have been taken into consideration in order to improve the quality of our manuscript.

No

Comments and Response

Page/Section/Location

1

"in vivo", "in vitro" and "via" should be written in italics throughout the text

In the revised manuscript, all occurrences of "in vivo," "in vitro," and "via" have been italic. Thank you.

Entire manuscript

2

"Gram-positive" and "Gram-negative" should be written with capital G.

Thank you so much for pointing out this inaccuracy. In the revised manuscript, the first letter of "Gram-positive" and "Gram-negative" has been written with capital G. Thank you.

Entire manuscript

3

The abbreviation "MD" from MD simulation should be explained where is first used in the text. 

I appreciate you bringing out this mistake. When the term "Molecular dynamics" (MD) was first used in the main text, we incorporated the abbreviation. Thank you.

Abstract

4

The Figures should be checked to ensure the writing is readable. For example, the color legend in Figure 2 is not readable.  

In the revised manuscript, figure 2's resolution was increased to make the letters more readable and noticeable. Thank you.

Figure 2

Round 2

Reviewer 1 Report

Dear Authors,

Thank you for all corrections. I recommend the manuscript for publication in the MDPI Journal. 

Minor comments:

I'm just not sure if 100% ethanol was actually used, wasn't it 96% ethanol?

Best Regards!

Author Response

POINT BY POINT RESPONSE TO REVIEWERS

Reviewer 1

Thank you for all corrections. I recommend the manuscript for publication in the MDPI Journal.

Thank you so much for recommending to publish our manuscript for publication in MDPI-Molecules Journal. Your suggestion has been taken into consideration in order to improve the quality of our manuscript.

No

Comments and Response

Page/Section/Location

1

I'm just not sure if 100% ethanol was actually used, wasn't it 96% ethanol?

I appreciate you bringing out this mistake. In the revised manuscript, we have made the necessary corrections and changed the ethanol concentration to 96%. Thank you.

Page 16, Section 4.3
